## How, in what contexts, and why do quality dashboards lead to improvements in care quality in acute hospitals? Protocol for a realist feasibility evaluation

Rebecca Randell [iD],[1,2] Natasha Alvarado,[2,3] Lynn McVey,[2,3] Joanne Greenhalgh,[4] Robert M West,[5] Amanda Farrin,[6] Chris Gale,[7] Roger Parslow,[7] Justin Keen [iD],[5] Mai Elshehaly,[8] Roy A Ruddle,[9] Julia Lake,[10] Mamas Mamas,[11] Richard Feltbower,[7] Dawn Dowding[12]

For numbered affiliations see end of article.

**Correspondence to**
Professor Rebecca Randell;
r.randell@bradford.ac.uk

## ABSTRACT

**Introduction** National audits are used to monitor care quality and safety and are anticipated to reduce unexplained variations in quality by stimulating quality improvement (QI). However, variation within and between providers in the extent of engagement with national audits means that the potential for national audit data to inform QI is not being realised. This study will undertake a feasibility evaluation of QualDash, a quality dashboard designed to support clinical teams and managers to explore data from two national audits, the Myocardial Ischaemia National Audit Project (MINAP) and the Paediatric Intensive Care Audit Network (PICANet).

**Methods and analysis** Realist evaluation, which involves building, testing and refining theories of how an intervention works, provides an overall framework for this feasibility study. Realist hypotheses that describe how, in what contexts, and why QualDash is expected to provide benefit will be tested across five hospitals. A controlled interrupted time series analysis, using key MINAP and PICANet measures, will provide preliminary evidence of the impact of QualDash, while ethnographic observations and interviews over 12 months will provide initial insight into contexts and mechanisms that lead to those impacts. Feasibility outcomes include the extent to which MINAP and PICANet data are used, data completeness in the audits, and the extent to which participants perceive QualDash to be useful and express the intention to continue using it after the study period.

**Ethics and dissemination** The study has been approved by the University of Leeds School of Healthcare Research Ethics Committee. Study results will provide an initial understanding of how, in what contexts, and why quality dashboards lead to improvements in care quality. These will be disseminated to academic audiences, study participants, hospital IT departments and national audits. If the results show a trial is feasible, we will disseminate the QualDash software through a stepped wedge cluster randomised trial.

### Strengths and limitations of this study

► This study will assess the feasibility of a trial of QualDash, a quality dashboard; if a trial is feasible, the findings will be used to inform the design of the definitive trial, determining the components of QualDash to be preserved, appropriate outcome measures and the contexts in which the trial should be undertaken.

► Through a controlled interrupted time series (CITS) study and qualitative multi-site case study, the study will also provide an initial understanding of whether the use of a quality dashboard leads to quality improvement, how, in what contexts, and why.

► The study will contribute to understanding of how realist methods can contribute to feasibility studies and the design of trials.

► Issues of data quality may be a limitation of the CITS study; data completeness, and whether this changes over the course of the study, will be assessed.

## INTRODUCTION

National clinical audits (NCAs), which provide comparative data on the performance of healthcare providers, are one of the means by which health systems around the world monitor care quality and safety. In England, a programme of over 30 NCAs is managed by the Healthcare Quality Improvement Partnership and all healthcare providers that contribute to delivery of the National Health Service (NHS) are required to participate. Such audits are anticipated to reduce unexplained variations in healthcare quality by stimulating quality improvement (QI).[1 2] While there is evidence of positive impacts of NCAs,[3–5] variation within and between providers in the extent to which they engage

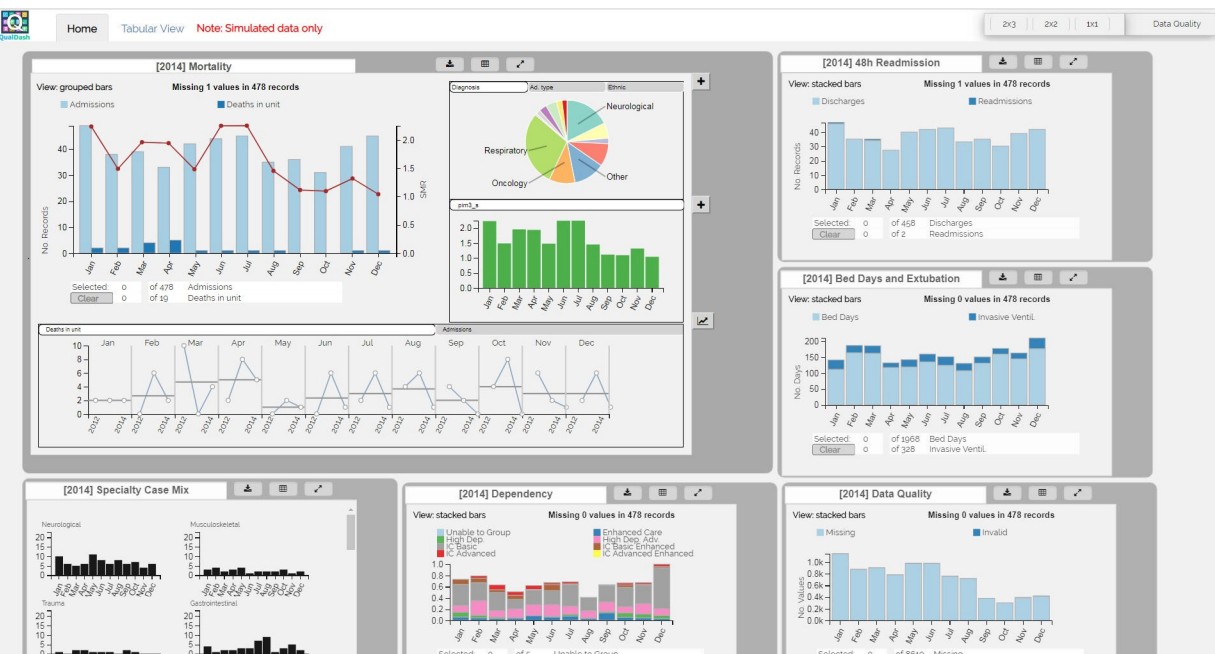

**Figure 1** Prototype of main dashboard view for the paediatric intensive care audit network (using simulated data).

with NCAs mean the potential for NCA data to inform QI is not being realised.[6 7]

Quality dashboards are a form of audit and feedback (A&F) that provide visualisations of audit data with the aim of informing QI efforts.[8] Healthcare providers are increasingly using quality dashboards. For example, quality dashboard use has been reported in Canada,[9] the UK[10] and the Netherlands.[11] While quality dashboards have been shown to have positive effects on some performance indicators,[9] empirical evidence regarding their impact remains limited.[12]

### QualDash
QualDash is an interactive web-based quality dashboard designed to support clinical teams and managers to explore data from two national audits, the Myocardial Ischaemia National Audit Project (MINAP) and the Paediatric Intensive Care Audit Network (PICANet), for the purpose of QI (figure 1). Information used to inform design of QualDash was collected through interviews with 50 clinicians and managers across five NHS Trusts (providers) and four healthcare commissioners, observations of meetings where audit data are discussed, a workshop with NCA suppliers, and two co-design workshops with clinicians and managers from one Trust.

The interviews revealed that use of NCA data is largely at the clinical team level, with more limited use at divisional and corporate (Board and sub-committees that report to the Board, such as Quality and Safety Committees) levels. At all levels, a key constraint in use of NCA data for QI is lack of access to timely data; there was consensus among interviewees that the data should not be more than 3 months old. QualDash seeks to improve access to timely data, providing users with a means to visualise the data they collect for the NCAs, without having to wait for data

to be returned to them from the NCAs. There is variation between Trusts in the extent to which NCA data are used, often related to resources, which in turn impacts on timeliness of data; Trusts that make greater use of NCA data tend to have local databases from which they can generate visualisations of the data (eg, bar charts) and audit support staff who have the time and skills to be able to generate such visualisations. In contrast, where such resources are not available, Trusts rely on the NCA annual reports, where data may be 15 months old (eg, one annual report published in June 2017 reported data from April 2015 to March 2016). QualDash provides visualisations of key metrics, each metric being represented within a 'QualCard' (figure 2), enabling Trusts to use NCA data for QI, regardless of existing resources. QualCards for MINAP and PICANet are listed in table 1; while there is only one set of QualCards for PICANet, for MINAP an additional QualCard is provided for teaching hospitals, as discussions with sites revealed that the metrics of interest are different between teaching hospitals and district general hospitals (DGHs). Sites are also able to create additional QualCards, to reflect local priorities.

To load new data into QualDash, NCA data are either extracted from the site's database or downloaded from the NCA website, and then fed to a small script (written in R), which in turn updates the dashboard. Users can add new data as often as they want, but at a minimum they will load data into QualDash at the same time as uploading to the NCAs (typically every 3 months).

The benefits perceived from using QualDash may vary between sites, with under-resourced sites that previously made little use of NCA data for QI perceiving greater impact than those that already have the means to use NCA data for QI. There are also constraints on use of NCA data

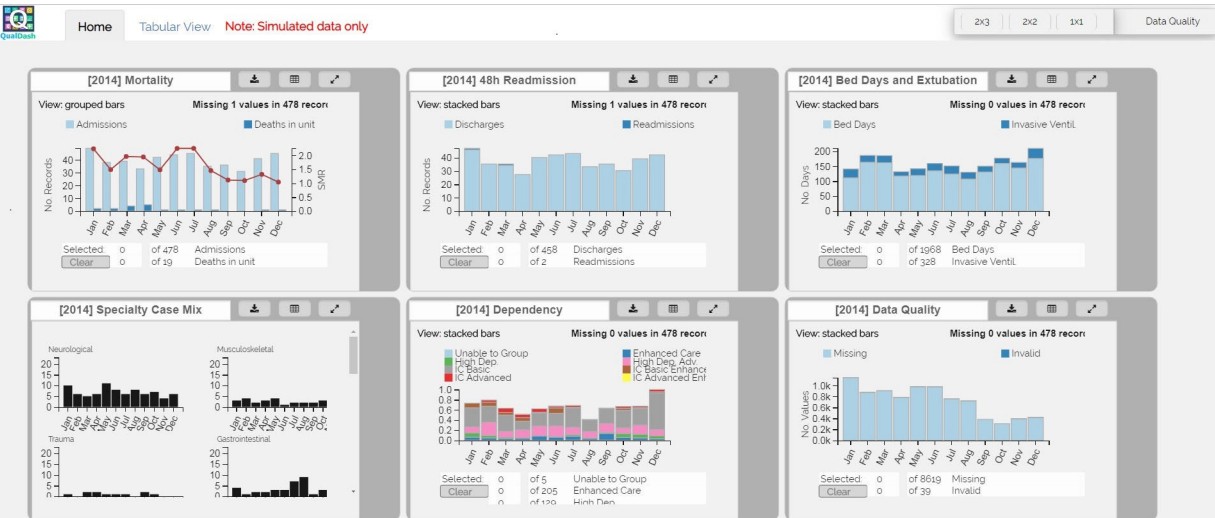

**Figure 2** Prototype of main dashboard view for the paediatric intensive care audit network with the mortality QualCard expanded (using simulated data).

for QI that it may be difficult for QualDash to address. For example, in some Trusts, clinical team members perceive that relevant managers will not agree to provide the resources necessary for QI initiatives, which reduces motivation to engage with NCA data and may affect the extent to which QualDash is used. However, QualDash provides means for visualisations to be downloaded and incorporated into presentations and reports, which may support clinical teams in making a stronger case for QI initiatives. Another constraint on use of NCA data for QI relates to clinicians' trust in the quality of the data. Interviews revealed variations across sites in processes for ensuring data quality. However, some interviewees also suggested that having the means to make more use of NCA data via QualDash would motivate them to improve their processes for ensuring data quality, although this will be dependent on local resources.

| Table 1 | QualCards |
| --- | --- |
| | **Metric** |
| MINAP—all sites | Mortality<br>Door (arrival in accident and emergency) to angiogram time<br>Gold standard drugs on discharge<br>Referral for cardiac rehabilitation<br>Acute use of aspirin |
| MINAP—teaching hospital specific | Call (by patient/relative to emergency services) to balloon (percutaneous coronary intervention) time |
| PICANet—all sites | Mortality<br>48 hours unplanned readmission<br>Bed days and accidental extubation<br>Specialty case mix<br>Data quality (number of records with a missing value)<br>Patient dependency |

In this paper, we describe the methods for a realist feasibility evaluation of QualDash. The study objectives are:
1. To develop an initial understanding of how, in what contexts, and why use of QualDash leads to QI; and
2. to assess the feasibility of conducting a trial of QualDash.

As no checklists exist for reporting of realist evaluation (RE) protocols, in presenting this protocol we draw on the RAMESES II reporting standards for REs[13] (online supplementary file 1).

## METHODS AND ANALYSIS
### Study design
The use of theory is needed for design and evaluation of A&F interventions,[14–16] and QI initiatives more generally.[17–19] This project draws on RE, which involves building, testing and refining theories about how an intervention is supposed to work.[20] These theories are expressed in the form of context, mechanism and outcome (CMO) configurations, where C+M=O, reflecting the realist understanding that it is recipients' responses to the resources that an intervention provides (the intervention mechanisms) that determine the impact of the intervention, and such responses are highly influenced by context.[21] Consequently, RE seeks to answer not only the question of 'what works?' but also 'what works for whom, in what circumstances, and why?'[22] It is concerned with both intended and unintended outcomes. RE is recommended for studying QI[23] and has been used for studying the implementation and impact of large-scale QI programmes.[24] There is increasing interest in use of realist methods in feasibility evaluations.[25–27]

We have drawn on a range of sources to develop CMO configurations which describe how, in what contexts, and why use of QualDash is anticipated to lead to QI (see online supplementary file 2). Data generated from the

interviews, observations and workshops described above have been essential to this, as have discussions with the designers of QualDash (ME and RAR) who, drawing on their expertise in information visualisation, have their own literature-informed theories regarding why certain features of QualDash will provide benefit to users.[28 29] We have also drawn on substantive theories regarding how A&F lead to QI at the micro,[30 31] meso[32] and macro levels.[33 34]

Data collection is designed to enable testing of the CMO configurations. Outcome data, in the form of key MINAP and PICANet measures, will be collected and analysed in a controlled interrupted time series (CITS) study, while a multi-site case study[35] will provide an initial understanding of the contexts and mechanisms that lead to those outcomes, as well as providing data on intermediate outcomes, such as increased use of NCA data. A&F interventions, and QI interventions more generally, require longitudinal evaluation to allow sufficient time for staff to implement changes and incorporate them into their practice.[36–38] Similarly, evaluation of health IT (HIT) should allow time for staff to integrate the technology into their practices and evolve those practices to take advantage of the functionality offered by the technology.[39] Therefore, data will be collected over a 12-month period, from August 2019.

### Public and patient involvement
A Lay Advisory Group has been established, which has contributed to the design of QualDash by reviewing the topic guide for the interviews that were conducted, providing their perspective on the findings of the interview study, and participating in the usability evaluation of QualDash. For the realist feasibility evaluation, they have provided advice on aspects to pay attention to when undertaking observations. They will contribute to analysis of a sample of the qualitative data, to provide a patient perspective. They will advise on dissemination of findings to relevant interest groups and will review outputs for comprehensibility.

### Setting/context
The feasibility study will be conducted in the five NHS acute Trusts in which the interview study that informed the design of QualDash was undertaken. Three Trusts are teaching hospitals that participate in both MINAP and PICANet and have been selected to ensure variation in key outcome measures (MINAP: 30-day mortality for patients hospitalised with ST-elevation myocardial infarction; PICANet: risk adjusted standardised mortality ratio). Two Trusts are DGHs that participate in MINAP but do not have a paediatric intensive care unit (PICU) and so do not participate in PICANet. These have been selected to ensure variation in the same key MINAP measure.

### Multi-site case study
In the multi-site case study, data will be collected through ethnographic observation and interviews. Ethnographic

methods have been argued as essential for studying implementation of QI interventions[19] and introduction of HIT.[40] Ethnography is well suited to RE because it involves observing phenomena in context, supporting understanding of how context influences the response to an intervention.[41] We will follow the Biography of Artefacts approach,[42] which is concerned with capturing how particular contexts and appropriations of a technology lead to different processes and generate different outcomes, a parallel to RE's concern with contexts, mechanisms, and outcomes.[43] It involves longitudinal 'strategic ethnography',[42] where data collection is guided by a provisional understanding of the moments and locales in which a technology and associated practices evolve.[43]

### Data collection
In the three teaching hospitals, we will undertake a minimum of 24 periods of observation per Trust, to be split across activities related to cardiology and the PICU, and in the two DGHs we will undertake a minimum of 12 periods of observation per Trust, to be spent observing activities related to cardiology. Each period of observation will be a minimum of 4 hours (total n=384 hours). While researchers will return to each Trust monthly, to understand how use of QualDash changes over time, more time will be spent in the first few months following the introduction of QualDash, because this is when users are most likely to engage with and explore the affordances of Qual-Dash and establish new practices around it, generating information with implications for system enhancement.[43] Observations will be scheduled to take place at different times of day and on different days of the week, to ensure the account of what is observed is as complete and representative as possible.[44]

At each case site, an initial phase of general observation will provide an opportunity for researchers to become familiar with the setting and for those in the setting to become familiar with the presence of the researchers. Following a previous study of dashboards,[10] observations will be undertaken in clinical areas to understand clinical teams' working practices and capture 'corridor committees' where issues of quality and safety are discussed more informally.[45] In the PICUs, initial observations will take place on the PICU, for example, with the researchers positioning themselves by the nurses' station, as well as observing handovers, safety huddles and ward rounds. Because activities related to cardiology tend to be more dispersed across hospitals, researchers will first shadow clinical team members (consultant cardiologists and acute chest pain nurses) to determine where it is most appropriate to conduct subsequent observations. These initial observations will also be used to record general details of the setting that may influence QualDash use, such as staffing levels and availability of computers.

After this initial phase, observation will be guided by the CMO configurations under investigation. In addition to observing formal meetings where quality and safety are discussed, predominantly at ward level but also at

divisional and corporate level, observation will involve shadowing staff members as they undertake particular activities: collection and entry of NCA data, to see if and how this changes over time; accessing and interrogating NCA data, whether using QualDash or some other means; preparation of reports and/or presentations using NCA data, again whether using QualDash or some other means. Where visualisations from QualDash are incorporated into presentations and written reports, we will follow the path of those documents, to identify staff members who may not use QualDash directly but are receiving QualDash outputs. Attention will be paid to how, in what contexts and why QualDash and QualDash outputs are used or not, understood in the context of broader practices and use of other sources of information for monitoring care quality, and how this changes over time. We will also follow local QI initiatives, recording data on, for example, when and how the need for the QI initiative was identified, contextual factors that appear to support and constrain its introduction, how the impact of the QI initiative is monitored, and other contextual factors that appear to influence the metric that the QI initiative is targeting. Researchers will record observations in fieldnotes, which will be written up in detail as soon after data collection as possible.

Brief interviews will be undertaken opportunistically during the course of conducting observations to clarify aspects of practice that are not immediately intelligible to an observer, with participant responses recorded in fieldnotes.[46] As data collection progresses, longer semi-structured interviews will be used to discuss revisions to our CMO configurations. These will be undertaken using a particular approach from RE, referred to as the teacher–learner cycle, whereby the theories under investigation are made explicit to the interviewee so that the interviewee can use their experiences to refine the researcher's understanding.[47] Being concerned with the reasoning of intervention recipients, mechanisms are often not observable,[21] so these longer interviews will also provide the opportunity to explore staff reasoning about QualDash. These longer interviews will be audio recorded and transcribed verbatim.

Logfiles are widely used to evaluate visualisation tools.[48] QualDash logfiles will record information about the user (job title and so on), data used (audit, year), overall time spent using QualDash, time spent interacting with different QualCards (including new QualCards that have been created), functionality used and whether QualDash visualisations were downloaded. In addition to providing data regarding extent of QualDash use, how QualDash is used and by whom, and how this changes over time, information from logfiles will be used to inform qualitative data collection (eg, asking in interviews why participants use particular QualCards and not others and the motivation behind the creation of new QualCards).

At the end of the data collection period, we will ask participants to complete a questionnaire based on the Technology Acceptance Model, using well-validated items that have been used in numerous evaluations of HIT,[49] including dashboards.[50] This will provide participants' perceptions of the usefulness of QualDash and data on whether they intend to continue using QualDash after the study period.

## Analysis

An iterative approach to data collection and analysis will be taken, to enable: ongoing testing and refinement of the CMO configurations; gathering of further data in light of such revisions; and refinement of QualDash in response to participants' feedback. Fieldnotes and interview transcripts will be entered into NVivo 11. Narrative analysis will be undertaken to develop a 'biography' of QualDash, which will describe use of QualDash and its outputs by a range of stakeholders at different levels (clinical team, divisional and corporate) and the interconnections between them.[10] Narrative analysis is consistent with a realist approach due to its emphasis on preserving connections within the data, thereby helping to understand causality.[51] This analysis will be supplemented with analysis of the logfiles and questionnaire data. Findings will be compared with the CMO configurations, to determine whether they support, refute or suggest a revision or addition to the CMO configurations.

## CITS study

Interrupted time series studies provide a robust method of assessing the effect of an intervention and have been used to assess effectiveness of a variety of complex interventions.[52] In a CITS, the addition of a control group enhances causal inference because the presence of seasonal trends and other potential time-varying confounders can be assessed.[53] Data will be collected across the five Trusts, with two control Trusts per intervention Trust, providing a total of 10 control Trusts. Control Trusts will be matched according to their size and outcomes pre intervention. Having more than one control site per intervention site increases power but, as the number of control sites per intervention site increases, quality of matching decreases. Therefore, we have chosen to have two control Trusts per intervention Trust to increase power while maintaining quality of the matching.

Given the study intention to determine the feasibility of and inform the design of a trial, a range of measures will be considered. Initially, we selected two process measures, one for MINAP and another for PICANet. For MINAP, we selected the composite process measure cumulative missed opportunities for care (CMOC). This has nine components (pre-hospital ECG, acute use of aspirin, timely perfusion, referral for cardiac rehabilitation and prescription at hospital discharge of what are considered to be the gold standard drugs—aspirin, thienopyridine inhibitor, angiotensin converting enzyme inhibitor, β-Hydroxy β-methylglutaryl-CoA (HMG-CoA) reductase inhibitor and beta blockers) and is inversely associated with mortality.[54] As some of these components, such as pre-hospital ECG, are outside the direct control of the

Trust, we will also explore the impact of QualDash on the individual measures that make up CMOC. On the basis of the measures that cardiology clinicians described in the interviews as being important for measuring care quality, we will also look at the percentage of patients who receive an angiogram within 72 hours from first admission to hospital, which is part of the Best Practice Tariff financial incentive scheme, and, for those hospitals that provide percutaneous coronary intervention (PCI), the proportion of patients who have a door-to-balloon time (the time from arrival at the hospital to PCI) of less than 60 min. Our CMO configurations (online supplementary file 2) suggest improvement will be seen in measures if: clinical teams perceive them as being important indicators of care and/or they relate to financial incentives; performance is not in line with expectations; they perceive the measure as being within their control; and the team is resourced to introduce QI initiatives in relation to these measures.

For PICANet, we selected use of non-invasive ventilation first for patients requiring ventilation, which has been shown to be associated with reduced mortality.[55] However, this was not raised as an area of concern in our interviews with PICU clinicians. On the basis of this and two additional considerations—it would require loading additional data into QualDash which would reduce the performance of QualDash in terms of speed and it requires computation of the data, while the focus of QualDash is on visualising the data—a QualCard has not been created for this metric. Therefore, while we will still include this measure in the CITS, we do not hypothesise that it will change, unless other sources of information, such as the PICANet annual report, draw a PICU team's attention to it. However, accidental extubation and unplanned readmission within 48 hours were identified in our interviews with PICU clinicians as being important indicators of care quality, so we will include these two measures in the CITS. On the basis of our CMO configurations (online supplementary file 2), we would expect to see an improvement in these measures in sites where performance is not in line with expectations, if the team is resourced to introduce QI initiatives in relation to these measures.

### Sample size considerations

A CITS study requires data for a minimum of three time points pre-intervention and three time points post-intervention and must also allow for any seasonal effect on the outcomes.[56] Monthly data will be obtained for 24 months pre-intervention and 12 months post-intervention. Consequently, for each intervention Trust, there will be 72 data points prior to introduction (24 for the intervention Trust and 48 for the control Trusts) and 36 data points post intervention (12 for the intervention Trust and 24 for the control Trusts). Sample size calculations were undertaken based on our two initial measures, CMOC for MINAP and use of non-invasive ventilation first for patients requiring ventilation for PICANet; full details are provided in online supplementary file 3.

### Analysis

Monthly MINAP and PICANet data will be extracted to spreadsheets for analysis with R software.[57] For both NCAs, each outcome will be regressed on time and the intervention. The time component will include a seasonal effect (quarterly effect) and will allow for a (linear) time trend. To account for clustering of monthly observations within hospitals, a random intercept will be fitted, although a fixed effect for hospital as a sensitivity analysis will be explored. Although the intervention is abrupt, its impact may well be 'phased in' over a few months, perhaps three. The timing of the bedding in of the intervention will be reported from the multi-site case study. Then a partial effect can be considered for this period with the interaction effect stepping up in a linear fashion.

The results of the CITS analysis will be incorporated into the biography of QualDash, the analysis of the data from the multi-site case study describing how contextual factors shape the evolution of practices around QualDash and how this leads to the resulting outcome pattern.

### Trial feasibility assessment and design

Our trial progression criteria are: (1) the number of people who engage with either MINAP or PICANet data (via QualDash or some other means) is the same or higher than the number of people who engaged with either MINAP or PICANet data prior to QualDash's introduction; (2) data completeness in the national audit improves or remains the same; (3) 50% or more of participants in the questionnaire survey perceive QualDash to be useful and express the intention to continue using it after the study period. Criteria (1) and (2) are concerned with ensuring the intervention does not have unintended negative consequences which would affect success of the intervention. Criterion (2) is also concerned with feasibility of outcome assessment. Criterion (3) is concerned with acceptability and uptake of the intervention, and therefore has implications for recruitment to a trial, as well as being concerned with participants' perceptions of the impact of QualDash on care. While not formally assessed as part of the progression criteria, the impact of QualDash on care as identified in the CITS will be considered in determining whether a future trial is justified. A traffic light system will be used to determine if a trial is feasible (green), feasible with modifications to QualDash (amber), or not feasible (red).[58 59]

If the results show a trial of QualDash is feasible, we will design a stepped wedge cluster randomised trial. Data from the CITS will be used to inform the selection of NCAs to be included in the trial (MINAP and/or PICANet) and will provide information about variability of outcomes and about how long a trial intervention period would need to be. Findings from the multi-site case study will be used to inform the selection of categories of user to be included in the trial and, associated with this, the level of randomisation (Trust, hospital or ward). Using the understanding of the relationship between contexts, mechanisms and outcomes provided by the study, we will

identify QualDash components associated with mechanisms that produce the desired outcomes in order for them to be preserved in the trial, while other components can be adapted to suit the local context.

## ETHICS AND DISSEMINATION

Written consent will be obtained from participants for interviews and for meeting observations.

Study results will provide initial understanding of how and in what contexts quality dashboards may lead to improvements in care quality. We will disseminate these results to academic audiences, study participants, hospital IT departments and NCAs. If we progress to a trial, in addition to providing further understanding of the impact of quality dashboards on care quality, this will result in wider dissemination of the QualDash software.

**Author affiliations**
¹Faculty of Health Studies, University of Bradford, Bradford, UK
²Wolfson Centre for Applied Health Research, Bradford, UK
³School of Healthcare, University of Leeds, Leeds, UK
⁴Sociology and Social Policy, University of Leeds, Leeds, UK
⁵Leeds Institute of Health Sciences, University of Leeds, Leeds, UK
⁶Clinical Trials Research Unit, University of Leeds, Leeds, UK
⁷School of Medicine, University of Leeds, Leeds, UK
⁸Faculty of Engineering & Informatics, University of Bradford, Bradford, UK
⁹School of Computing, University of Leeds, Leeds, UK
¹⁰Leeds Teaching Hospitals NHS Trust, Leeds, UK
¹¹Royal Stoke University Hospital, Stoke-on-Trent, UK
¹²School of Health Sciences, University of Manchester, Manchester, UK

**Contributors** RR is principal investigator for the study, she conceived, designed and secured funding for the study in collaboration with JG, RMW, AF, CG, RP, JK, RAR, JL, MM and DD. NA and LM led the qualitative data collection and analysis that informed the design of QualDash and the design of evaluation. ME developed the QualDash software and contributed to the design of the evaluation. RP and RF provided data for the testing of QualDash and provided significant feedback on its design. All authors provided input into various aspects of the evaluation design and revised drafts of the protocol. RR led the writing of this protocol manuscript. All authors read and approved the final manuscript.

**Funding** This research is funded by the National Institute for Health Research (NIHR) Health Services and Delivery Research (HS&DR) Programme (project number 16/04/06). The views and opinions expressed are those of the presenter and do not necessarily reflect those of the HS&DR programme, NIHR, NHS or the Department of Health.

**Competing interests** CG is a member of the MINAP Academic and Steering Groups. RF is the principal investigator for PICANet and RP was previously Principal Investigator for PICANet.

**Patient consent for publication** Not required.

**Ethics approval** Ethics approval has been received from the University of Leeds School of Healthcare Research Ethics Committee (Approval no.HREC16-044).

**Provenance and peer review** Not commissioned; externally peer reviewed.

**ORCID iDs**
Rebecca Randell http://orcid.org/0000-0002-5856-4912
Justin Keen http://orcid.org/0000-0003-2753-8276

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
