## [Reviewer comments · BMJ Open]

ARTICLE DETAILS

TITLE (PROVISIONAL)	How, in what contexts, and why do quality dashboards lead to improvements in care quality in acute hospitals? Protocol for a realist feasibility evaluation
AUTHORS	Randell, Rebecca; Alvarado, Natasha; McVey, Lynn; Greenhalgh, Joanne; West, Robert; Farrin, Amanda; Gale, Chris; Parslow, Roger; Keen, Justin; Elshehaly, Mai; Ruddle, Roy; Lake, Julia; Mamas, Mamas; Feltbower, Richard; Dowding, Dawn

VERSION 1 – REVIEW

REVIEWER	Professor Clair Sullivan University of Queensland Queensland University of Technology Brisbane, Queensland, Australia
REVIEW RETURNED	24-Aug-2019

GENERAL COMMENTS	Thank you for this interesting paper. It is well written and has a clear vision for assessing the impact of the introduction of clinical analytics into healthcare settings. Evidence based guidance on the effectiveness of the introduction of dashboards to support quality improvement in healthcare is sparse in the literature so I want to heartily encourage work. This is quite a speculative paper, really describing the methods that might be used to assess the impact (on clinical outcomes) of the introduction of clinical dashboards. A few comments 1.it is unclear what the product being introduced actually is and whether it is uniform across the sites? A critical factor for clinicians to engage successfully with analytics is the frequency of the data refresh. It is unclear if the registry data are drawn from live electronic medical records (EMRs) or require retrospective data entry by coders or clinicians, so the provenance and age of the data presented to the clinicians is hard to discern from this paper as written. Is QualDash a live stream from a data warehouse or is manual extraction into a flat file with a business intelligence presentation layer over the top ? If Qualdash does not provide live streaming analytics how long does it take from the clinical event occurring to the event data being aggregated in the dashboard and presented to the clinicians ?
--

	Evidence shows that live streaming analytics are more effective than static, retrospective reports or “flat” dashboards that are updated infrequently with retrospective data. Some variation is noted in this paper in the data warehousing capability of the sites, are they all able to present data with the same frequency of refresh? A feed with 15 min updates is likely to have more engaged clinicians than retrospective reports produced monthly. Is the time to presentation of data consistent across all the data elements and the sites? If not, how will this be accounted for in the analysis? 2. Quality improvement methods deployed It is unclear if there are governance and quality improvement resourcing and methodologies arranged around QualDash. I imagine that the qualitative evaluation is part of this assessment. It will be challenging to combine the multifaceted impact of quality improvement method, resourcing, governance, and data presentation variables to achieve a globally meaningful result. How will these qualitative and quantitative results be combined, presented and analysed meaningfully? 3. Statistical analysis Attempts at RCTs or matched studies for complex health IT implementations are notoriously difficult due to multiple confounders and issues. I note the number of data points sampled is mentioned in this paper, but no power calculation is performed. The primary end point of the study is not clearly defined and it is not articulated if the study is powered to reach this endpoint. That does not mean we should not try and analyse this important data, however I would recommend a pilot study.. Undertake a pre post study on a single site (acknowledging that unfortunately there may be confounders introduced during the study at the single site), Define the product clearly (informatics such as veracity of data, age of data, method of presentation etc) and the primary end point, define and exclude a wash-in period during adoption and optimisation from the analysis , calculate power needed for endpoint and then compare a similar period pre and post implementation matched for time and activity. This simpler analysis will allow a deep understanding of the duration of the washin period for adoption and optimisation of the product. This quantitative method once perfected, can be scaled to other sites. Qualitative analysis should proceed to provide the important context for the outcomes measured. Combining the two methods would give a rich and powerful understanding of the context and products that are useful. Thank you for your work to evaluate the introduction of clinical analytics into healthcare settings. It is undoubtedly the future of quality improvement. I look forward to seeing the results of the analysis
--	---

REVIEWER	Dr Rebecca Ann Simms
-----------------	----------------------

	University Hospitals Bristol NHS Foundation Trust, UK and University of Bristol, UK
REVIEW RETURNED	06-Sep-2019

GENERAL COMMENTS	This is an extremely thorough and well thought out protocol for a feasibility evaluation of a quality dashboard and its implementation across multiple NHS hospital Trusts. The mixed methods approach will likely lead to a rich range of data with potential answers to the 'how' and 'why' rather than just 'if' the use of a quality dashboard will improve set outcome measures. This will likely be of intense interest to those researchers and clinicians involved in improvement science across a range of healthcare settings. There were just 2 small points I wished to raise with the authors:  1. Has the findings from the preliminary research work used to inform the design of the QualDash been published (or planned for submission) anywhere? This will likely be of interest to those researchers designing quality dashboards and more broadly, researchers and clinicians wishing to understand the potential constraints and barriers to national and local quality monitoring initiatives. 2. There appears to be no mention of the source of the data inputs for the QualDash programme. Are these extracted from HES data, local/national databases or other sources? The potential inaccuracies in these data sources should be discussed and considered as they may represent limitations in the QualDash outputs and subsequent actions from these. I look forward to seeing the results of this proposed study once completed.
---

REVIEWER	Martine de Bruijne EMGO Institute, VU University Medical Center, Amsterdam, the Netherlands, Dept. of Public and Occupational Health
REVIEW RETURNED	15-Sep-2019

GENERAL COMMENTS	Thank you for the opportunity to review this manuscript. To my opinion it is well written, clearly structured and very informative for researchers who want to undertake similar research. Methods are well explained and linked to literature on theory and examples. Moreover, I highly estimate the transparent report on the current context on the choice of indicators for the CITS. On page 7 line 14-25 you shortly describe the development of CMO configurations in file 2. This could be a paper by its own. Lessons that can be learned from this preparatory phase reach much further than the information in additional file 2. It would especially be interesting to analyse interrelations between the individual CMO statements. Also negative effects can be identified and analysed. It would be very interesting to add this to the paper. On page 12 line 5 it is stated that two Control Trusts will be compared to five intervention trusts. Why this disbalance? I would assume the sources of variation in both groups would be the same, thus a balanced comparison would be more logical. I miss a short discussion of expected results and the strengths and limitations of the study design you are aware of or anticipating.
--

VERSION 1 – AUTHOR RESPONSE

Reviewer 1

“It is unclear what the product being introduced actually is and whether it is uniform across the sites?”
Two images are provided to illustrate the interface of QualDash. We have now added a table (Table 1) which specifies the QualCards that are provided and where there is variation between sites. As specified in the description of QualDash, there is also the potential for users to create additional QualCards. Through the logfiles, we will track the extent to which new QualCards are created and interacted with; we have added this to the description of the information captured by the logfiles.

“It is unclear if the registry data are drawn from live electronic medical records (EMRs) or require retrospective data entry by coders or clinicians, so the provenance and age of the data presented to the clinicians is hard to discern from this paper as written.”

In all sites and for both audits, data are recorded on paper forms and then entered into either a local database (if they have one) or directly into the supplier database. This process is determined by the audits and is outside of the control of QualDash. The study is a response to a commissioned call for research into the presentation of NHS audit data to clinical teams for the purpose of quality improvement; how the data are gathered was outside the scope of the call, although we acknowledge that these processes do impact the timeliness of the data that QualDash is able to present. Sites will load data into QualDash at least every three months but can upload data as frequently as they want to; we have added a sentence to the paper to explain this, along with an explanation of how the data are loaded into QualDash.

“Is QualDash a live stream from a data warehouse or is manual extraction into a flat file with a business intelligence presentation layer over the top? If Qualdash does not provide live streaming analytics how long does it take from the clinical event occurring to the event data being aggregated in the dashboard and presented to the clinicians? Evidence shows that live streaming analytics are more effective than static, retrospective reports or “flat” dashboards that are updated infrequently with retrospective data. Some variation is noted in this paper in the data warehousing capability of the sites, are they all able to present data with the same frequency of refresh? A feed with 15 min updates is likely to have more engaged clinicians than retrospective reports produced monthly. Is the time to presentation of data consistent across all the data elements and the sites? If not, how will this be accounted for in the analysis?”

As explained above, QualDash is not a live stream from a data warehouse. We have an automated process for data extraction (an R script) so it is not a manual process. We ask someone on-site to run the script; therefore it can be described as a semi-automatic extraction. Some sites have opted to run a scheduled process for the R script, in which case the process of feeding new data into QualDash would be fully automated at those sites. For the presentation part, it could be described as a business intelligence layer, given that dashboards fall under the definition of business intelligence, but, unlike static dashboards, QualDash comprises an engine for automatic dashboard generation based on configuration files that are co-designed and customised to meet end users' specific needs.

The time between a clinical event occurring and it being presented to clinicians will vary according to how frequently staff upload the data. Through the observations, we will gather information about the process of uploading data to QualDash, how this varies between sites, and if and how this changes over the period of the evaluation; for example, it could be anticipated that, being able to make use of the data for quality improvement, staff are motivated to upload the data more frequently, as captured in our first CMO configuration. At one site, they are discussing uploading data on a weekly basis.

“It is unclear if there are governance and quality improvement resourcing and methodologies arranged around QualDash. I imagine that the qualitative evaluation is part of this assessment. It will be challenging to combine the multifaceted impact of quality improvement method, resourcing, governance, and data presentation variables to achieve a globally meaningful result. How will these qualitative and quantitative results be combined, presented and analysed meaningfully?”

Our interviews revealed that sites vary in terms of the quality improvement processes they have in place and the resources provided to support quality improvement. We anticipate that these contextual factors will impact their use of QualDash, as captured in our CMO configurations (particularly 2, 5, and 6). We have chosen to use realist evaluation for this study because it is an explicitly mixed method approach that will allow us to combine, through the testing of the CMO configurations, qualitative and quantitative data in order to understand how context influences the impacts of QualDash.

“Attempts at RCTs or matched studies for complex health IT implementations are notoriously difficult due to multiple confounders and issues. I note the number of data points sampled is mentioned in this paper, but no power calculation is performed. The primary end point of the study is not clearly defined and it is not articulated if the study is powered to reach this endpoint. That does not mean we should not try and analyse this important data, however I would recommend a pilot study. Undertake a pre post study on a single site (acknowledging that unfortunately there may be confounders introduced during the study at the single site).”

We have now added information about the sample size calculation for the CITS; due to space limitations, we have placed this in Additional file 3. A number of the points that the reviewer raises reflect the fact that the study is a feasibility study, intended to assess the feasibility of a full trial and, if a trial is deemed to be feasible, to inform the design of that trial. We are intentionally considering a range of outcomes in the CITS in order to determine what might be an appropriate outcome for a trial. We have chosen to undertake the feasibility study across multiple sites to allow an understanding of the impact of context prior to wider roll out in a trial. We have reordered the text to make it clearer how the findings are intended to inform a future trial and have added further details regarding this.

“Define the product clearly (informatics such as veracity of data, age of data, method of presentation etc) and the primary end point, define and exclude a wash-in period during adoption and optimisation from the analysis, calculate power needed for endpoint and then compare a similar period pre and post implementation matched for time and activity.”

A number of the points here would be appropriate if we were not undertaking a feasibility study. We have explained above why we are considering a range of outcomes. We agree that defining a wash-in period is important. However, given that this is a feasibility study, it is not appropriate to define this in advance as we do not have data that would allow us to make a decision about the duration of the wash-in period. As stated in the protocol, the qualitative data will be used to determine the wash-in period for the CITS (we describe this as the impact being ‘phased in’).

Reviewer 2

“Has the findings from the preliminary research work used to inform the design of the QualDash been published (or planned for submission) anywhere?”

We plan to publish two journal papers on the interview study that informed the design of QualDash; one is currently under review, while the other is currently in preparation.

“There appears to be no mention of the source of the data inputs for the QualDash programme. Are these extracted from HES data, local/national databases or other sources? The potential inaccuracies in these data sources should be discussed and considered as they may represent limitations in the QualDash outputs and subsequent actions from these.”

The reviewer raises an important point. As described above, in all our sites and for both audits, data are initially collected on paper forms. In the description of QualDash, we have acknowledged trust in data quality as a constraint on use of NCA data which may impact use of QualDash. However, it may be that being able to use QualDash makes staff more aware of inaccuracies in the data (QualDash shows the number of records where information related to a particular metric is incomplete) and motivates them to improve their processes for ensuring data quality, a hypothesis that is captured in our first CMO configuration. In the strengths and limitations section, we have added issues of data quality as a potential methodological limitation of the CITS.

Reviewer 3

“On page 7 line 14-25 you shortly describe the development of CMO configurations in file 2. This could be a paper by its own. Lessons that can be learned from this preparatory phase reach much further than the information in additional file 2. It would especially be interesting to analyse interrelations between the individual CMO statements. Also negative effects can be identified and analysed. It would be very interesting to add this to the paper.”

Due to word limitations, we are not able to add this to the paper. However, the findings of the interview study will be reported elsewhere.

“On page 12 line 5 it is stated that two Control Trusts will be compared to five intervention trusts. Why this disbalance? I would assume the sources of variation in both groups would be the same, thus a balanced comparison would be more logical.”

There will be two control Trusts per intervention Trust, so five intervention Trusts and ten control Trusts. Having more than one control Trust per intervention Trust increases power but, as the number of control Trusts per intervention Trust increases, the quality of matching decreases. Therefore, we have chosen to have two control Trusts per intervention Trust to increase power while maintaining the quality of the matching. We have edited the text to make the total number of control Trusts clearer and to explain our reasoning behind the choice.

"I miss a short discussion of expected results and the strengths and limitations of the study design you are aware of or anticipating."

Strengths of the study design are described in the 'Strengths and limitations' section; because of word limits, we have not repeated these in the body of the paper. We have revised this section to include methodological limitations. The expected results – or the study hypotheses – are captured in the CMO configurations. Again, we have not repeated these due to word limits.

VERSION 2 – REVIEW

REVIEWER	Clair Sullivan Royal Brisbane Hospital Metro North Hospital and Health Service university of Queensland Queensland University of Technology
REVIEW RETURNED	31-Oct-2019

GENERAL COMMENTS	Thanks for the revisions. The issues with data age and validity have been well-addressed. A few suggestions: The objective of this paper needs to be clearer. The abstract describes this is a feasibility evaluation of Qualdash. It is not really clear to the educated reader that this initial study is really just to work out the protocol for the evaluation, it reads as though this could be the evaluation proper. eg. "This study combines a controlled interrupted time series study with a qualitative multi-site case study in order to provide an understanding of not only whether use of a quality dashboard leads to quality improvement but also how, in what contexts, and why. In addition to assessing the feasibility of a trial, the study will determine the components of QualDash to be preserved in a definitive trial, appropriate outcome measures, and the contexts in which a definitive trial should be undertaken. It is a little confusing whether this is the evaluation proper or just the feasibility study for the evaluation when reading the abstract alone. This paper is a bit of both, which is fine but just be clear about that in abstract. It is defined in the introduction as two questions eg. "In this paper, we describe the methods for a realist feasibility evaluation of QualDash. The study objectives are:
--

	1. To understand how, in what contexts, and why use of QualDash leads to QI; and 2. To assess the feasibility of conducting a trial of QualDash" It is important that the data quality and age issues are addressed when developing controls. How will this be approached? eg. data that is entered every 6 mths may have a different momentum and relevance to the clinical teams than data that is entered weekly?
--	---

REVIEWER	Rebecca Simms University Hospitals Bristol NHS Foundation Trust and University of Bristol, UK
REVIEW RETURNED	26-Oct-2019

GENERAL COMMENTS	Thank you for submitting a revised version of your protocol paper in view of our review comments. This changes have improved your paper and added more clarity to the work that you have already completed and the details of your planned realist feasibility evaluation. There is one point, however, that I would like to see some clarification or amendment for. On page 13 you state that 'sample size calculations were undertaken based on our two initial measures, CMOC for MINAP and use of non-invasive ventilation first for patients requiring ventilation for PICANet'. However, as the PICANet process measure initially selected (non-invasive ventilation first for patients requiring ventilation) was later opted not to be used in the study, please can you explain why you have not repeated the sample size calculations using your new PICANet measures? I wish you well with your intended study and I look forward to reading your results upon publication.
--

REVIEWER	Martine de Bruijne EMGO Institute, VU University Medical Center, Amsterdam, the Netherlands, Dept. of Public and Occupational Health
REVIEW RETURNED	03-Dec-2019

GENERAL COMMENTS	Thank you for your adequate responses to the reviewer's comments.
---

VERSION 2 – AUTHOR RESPONSE

Response to reviewers' comments

Reviewer 1

"The objective of this paper needs to be clearer. The abstract describes this is a feasibility evaluation of Qualdash. It is not really clear to the educated reader that this initial study is really just to work out the protocol for the evaluation, it reads as though this could be the evaluation proper."

As the reviewer points out, we have two objectives. However, we acknowledge that this could be confusing for the reader. We have therefore made revisions at several points throughout the paper – in the abstract, strengths and limitations, objectives, study design, and setting/context – to emphasise that it is a feasibility study and that the study will provide an initial, rather than definitive, understanding of how, in what contexts, and why use of QualDash leads to quality improvement.

“It is important that the data quality and age issues are addressed when developing controls. How will this be approached? eg. data that is entered every 6 mths may have a different momentum and relevance to the clinical teams than data that is entered weekly?”

Information is not available regarding age of data to inform our choice of control sites. However, we acknowledge that this is an important issue and we will be able to draw on our knowledge regarding how frequently the data are updated, gathered through the qualitative data collection, when interpreting differences in impacts across the intervention sites.

Reviewer 2

“There is one point, however, that I would like to see some clarification or amendment for. On page 13 you state that 'sample size calculations were undertaken based on our two initial measures, CMOC for MINAP and use of non-invasive ventilation first for patients requiring ventilation for PICANet'. However, as the PICANet process measure initially selected (non-invasive ventilation first for patients requiring ventilation) was later opted not to be used in the study, please can you explain why you have not repeated the sample size calculations using your new PICANet measures?”

We are still using this process measure in the study (although we hypothesise that there will not be a change in this measure). However, we acknowledge that this was not adequately clear and so we have revised the wording to make this more explicit.

VERSION 3 – REVIEW

REVIEWER	Clair Sullivan University of Queensland Australia
REVIEW RETURNED	28-Dec-2019

GENERAL COMMENTS	Thanks for the revision . It reads better. In the article summary, it would be worth outlining more clearly that this study is a preliminary assessment to inform a definitive trial. There is too much emphasis on the method (realist evaluation discussed in detail) and perhaps not quite enough on the planned outcomes (definitive trial design).
---

	Data quality (age, completeness, accuracy) is likely to vary across the trusts as we have previously discussed . Eg a trust using real time data from a data warehouse may have deep engagement with the dashboards whereas a trust using 6 mth old incomplete data sets may struggle to gain traction. Will your matching process for controls accomodate this? Minor comments p4 Please define CITS
--	--

REVIEWER	Dr Rebecca Simms University Hospitals Bristol NHS Foundation Trust and University of Bristol
REVIEW RETURNED	05-Jan-2020

GENERAL COMMENTS	Thank you for your clarification of the previously mentioned reviewer points. These have now been fully addressed and the paper has been improved subsequently.
---

VERSION 3 – AUTHOR RESPONSE

Response to reviewer comments

Reviewer 1

“In the article summary, it would be worth outlining more clearly that this study is a preliminary assessment to inform a definitive trial. There is too much emphasis on the method (realist evaluation discussed in detail) and perhaps not quite enough on the planned outcomes (definitive trial design).” We have reordered the bullet points in the article summary, to place the emphasis on assessment of trial feasibility and design of a definitive trial. We have also changed the language to make clearer that the study will inform the design of a definitive trial.

“Data quality (age, completeness, accuracy) is likely to vary across the trusts as we have previously discussed . Eg a trust using real time data from a data warehouse may have deep engagement with the dashboards whereas a trust using 6 mth old incomplete data sets may struggle to gain traction. Will your matching process for controls accomodate this?”

As we noted in our previous response, information is not available regarding age of data to inform our choice of control sites. Both MINAP and PICANet do publish some information regarding data completeness in terms of case ascertainment. However, for PICANet, case ascertainment is determined via a validation visit and not every PICU is visited each year, so that these data are not available for all sites. For MINAP, case ascertainment is determined through comparing the number of cases submitted to MINAP with the number of cases in the Hospital Episode Statistics (HES) data that are coded as myocardial infarction. The latest MINAP report points to some hospitals submitting many more cases to MINAP than appear in HES, suggesting differences in hospital coding practices. Consequently, we do not consider this as a reliable source of data on which to base our matching. More generally, we have discussed the issue you raise with our statistician who has advised that matching should be done on a small number of important aspects, hence the decision to focus on size and preintervention outcomes, and that to match on additional features would not be feasible.

“p4 Please define CITS”

Thank you for pointing out this oversight. We have now addressed this.

VERSION 4 – REVIEW

REVIEWER	
REVIEW RETURNED	

GENERAL COMMENTS	
--

REVIEWER	
REVIEW RETURNED	

GENERAL COMMENTS	
--

REVIEWER	
REVIEW RETURNED	

GENERAL COMMENTS	
--

REVIEWER	
REVIEW RETURNED	

GENERAL COMMENTS	
--

REVIEWER	
REVIEW RETURNED	

GENERAL COMMENTS	
--

VERSION 4 – AUTHOR RESPONSE

VERSION 5 – REVIEW

REVIEWER	
REVIEW RETURNED	

GENERAL COMMENTS	
--

REVIEWER	
REVIEW RETURNED	

GENERAL COMMENTS	
--

REVIEWER	
REVIEW RETURNED	

GENERAL COMMENTS	
--

REVIEWER	
REVIEW RETURNED	

GENERAL COMMENTS	
--

REVIEWER	
REVIEW RETURNED	
GENERAL COMMENTS	

VERSION 5 – AUTHOR RESPONSE